# Diagnosis of Inherited Platelet Disorders: Clinical Evaluation and Functional and Molecular Assays

**DOI:** 10.3390/biom15060846

**Published:** 2025-06-10

**Authors:** Ana Sánchez-Fuentes, Juliana Pérez-Botero, José M. Bastida, José Rivera

**Affiliations:** 1Servicio de Hematología, Hospital Universitario Morales Meseguer, Centro Regional de Hemodonación, Instituto Murciano de Investigaciones Biomédicas (IMIB)-Pascual Parrilla, Universidad de Murcia, CIBERER-ISCIII-U765, 30503 Murcia, Spain; ana.sanchez1610@gmail.com; 2Division of Hematopathology, Mayo Clinic, Rochester, MN 55905, USA; perezbotero.juliana@mayo.edu; 3Departamento de Hematología, Complejo Asistencial Universitario de Salamanca (CAUSA), Instituto de Investigación Biomédica de Salamanca (IBSAL), Centro de Investigación del Cáncer, IBMCC-CSIC, Universidad de Salamanca (USAL), 37007 Salamanca, Spain; chema@usal.es; 4Coordinators of Grupo Español de Alteraciones Plaquetarias Congénitas (GEAPC) of SETH c/Aravaca 12, 28040 Madrid, Spain

**Keywords:** inherited platelet disorders, inherited thrombocytopenia, inherited platelet function disorder, platelet function testing, IPD genetic diagnosis

## Abstract

Inherited platelet disorders (IPDs) are a group of rare conditions affecting platelet number, function, or both. Clinical manifestations vary widely, from asymptomatic cases to patients with severe bleeding, syndromic features, or early-onset blood cancers. Some are diagnosed due to family history. Early and accurate diagnosis—through both clinical and molecular evaluation—is essential for effective patient management and to avoid inappropriate treatments due to misdiagnosis. Genetic confirmation aids in prognosis, follow-up planning, family screening, genetic counseling, and donor selection for stem cell transplantation if required. However, diagnosing IPD is still challenging due to the disorders’ complexity and the limitations of current lab tests. This review outlines the diagnostic process for IPD and provides evidence-based, practical recommendations informed by scientific literature and clinical experience.

## 1. Background and Objectives

Inherited platelet disorders (IPD) are a heterogeneous group of rare diseases in which the number of platelets in the blood, inherited thrombocytopenias (IT), the platelet function (inherited disorders of platelet function [IPFD]), or both parameters are affected simultaneously. The clinical presentation of patients is highly variable and includes a wide range of symptoms and laboratory abnormalities. Asymptomatic patients with mild abnormalities may be identified during routine studies (e.g., moderate thrombocytopenia in the complete blood count). Other patients present with moderate–severe or even life-threatening bleeding, severe syndromes affecting different organs and tissues, and others with a high risk of developing hematological malignancies at an early age [1,2,3,4]. Some patients with an IPD are evaluated due to a family history of bleeding, thrombocytopenia, or another specific platelet disorder.

The importance of an early and accurate diagnosis of ITP, combining phenotypic and molecular characterization, is essential for the correct clinical management of patients. Historically, around 30% of patients with an IPD are diagnosed in adulthood and 8–15% receive unnecessary treatments for immune thrombocytopenia (ITP), such as corticosteroids and/or splenectomy, among others [5,6]. Furthermore, confirming the genetic diagnosis helps establish the prognosis, plan for specific follow-up, facilitate genetic counseling and family screening, and is critical in donor selection if the patient requires a hematopoietic progenitor cell transplant [1].

The accurate diagnosis of platelet function defect (IPFD) represents a significant challenge due to the clinical and molecular heterogeneity of these disorders, as well as the limited reproducibility and specificity of currently available platelet function tests [1,7,8,9,10,11]. It requires a comprehensive evaluation of the clinical history and functional and molecular testing which, if possible, should be performed in reference centers capable of performing and interpreting the indicated tests.

This review addresses the different stages of the IPD diagnostic process (Figure 1), and aims to offer practical recommendations, based on current scientific evidence and our own experience.

## 2. Personal and Family History of the Patients with Suspected IPD

### 2.1. Personal History of Patients with Suspected IPD

The first step in the diagnostic process for an IPD is a thorough assessment of personal and family medical history, based on two guiding symptoms: bleeding and/or thrombocytopenia. The focus should also be on ruling out the pathologies most frequently associated with these symptoms (ITP, autoimmune diseases, liver disease, uremia, other hemorrhagic coagulopathies) or the use of medications and/or dietary supplements [12].

After potential acquired causes of platelet disease have been ruled out, the clinical picture typically reflects one of the following scenarios.

#### 2.1.1. The Patients with Bleeding With or Without Thrombocytopenia

The main consideration in this patient population is that bleeding is excessive and secondary to impaired platelet function. The clinical evaluation is typically approached as follows:▪A physical examination of the patient, looking for external signs of bleeding.▪Evaluate whether the patient’s current and/or previous bleeding symptoms meet the typical characteristics of a primary hemostatic disorder: mucocutaneous bleeding, immediate bleeding after trauma or in the immediate postoperative period. It should be noted that these signs are also typical of other disorders such as von Willebrand disease (VWD) or endothelium-related pathology. In some severe IPFDs, such as Glanzmann thrombasthenia (GT), there may also be urinary, muscular bleeding and hemarthrosis, which are generally more typical of secondary hemostatic disorders such as hemophilia.▪Objectively assess bleeding using validated tools such as the bleeding assessment tool developed by the International Society on Thrombosis and Hemostasis (ISTH-BAT) [13,14,15] (Table 1). The absolute value of the ISTH-BAT (bleeding score) that is considered pathological, and therefore, suggestive of platelet dysfunction is ≥3 points in children, ≥4 in men, and ≥6 in women. An ISTH-BAT ≥ 6 has been described to have a positive predictive value of 99% for an IPFD [15], but a lower score does not exclude the existence of a possible mild or moderate IPFD. Its main limitations are the pediatric population and/or young adults without hemostatic challenges, recall bias, the saturation of the score with recurrent symptoms, and its inability to distinguish different types of bleeding disorders.▪In addition to the severity of bleeding in terms of the absolute ISTH-BAT value, it is also important to assess the age of onset, persistence, and location of bleeding.▪In cases of patients with thrombocytopenia, concomitant platelet dysfunction is suspected if bleeding is disproportionate to the patient’s platelet count. In other systemic diseases or in ITP, spontaneous or even severe bleeding is usually not observed if the platelet count exceeds 10 × 10^9^/L [16,17].▪In some IPDs with mild impairment of platelet function, bleeding is not an essential feature, and a lack of bleeding manifestations does not rule out the diagnosis.

**Table 1 biomolecules-15-00846-t001:** Bleeding score according to the International Society of Thrombosis and Hemostasis Bleeding Assessment Tool (ISTH-BAT) [18].

Symptom	Score				
	0	1	2	3	4
Epistaxis	No/Trivial	>5/Year or >10 min	Consultation only ^a^	Packing or cauterization or antifibrinolytic	BT/RT/DDAVP
Cutaneous	No/Trivial	For bruises 5 or more [>1 cm] in exposed areas	Consultation only ^a^	Extensive	Spontaneous hematoma requiring BT
Bleeding from Minor Wounds	No/Trivial	>5/y or more than 10 min	Consultation only ^a^	Surgical hemostasis	BT/RT/DDAVP
Oral Cavity	No/Trivial	Present	Consultation only ^a^	Surgical hemostasis or antifibrinolytic	BT/RT/DDAVP
GI Bleeding	No/Trivial	Present (not associated with GI lesions)	Consultation only ^a^	Surgical hemostasis or antifibrinolytic	BT/RT/DDAVP
Hematuria	No/Trivial	Present (macroscopic)	Consultation only ^a^	Surgical hemostasis iron therapy	BT/RT/DDAVP
Tooth Extractions	No/Trivial or not carried out	Reported in ≤25% of all procedures, no interventions	Reported in >25% of procedures, no interventions	Resuturing or packing	BT/RT/DDAVP
Surgery	No/Trivial or not carried out	Reported in ≤25% of all procedures, no interventions	Reported in >25% of procedures, no interventions	Surgical hemostasis or antifibrinolytic	BT/RT/DDAVP
Menorrhagia	No/Trivial	Consultation only ^a^ or hanging pads more frequently than every 2 h; or clot sand flooding; or PBAC score > 100	Time of work/school > 2/y or requiring antifibrinolytic or hormonal therapy or iron therapy	Requiring combined treatment with antifibrinolytic agents and hormonal therapy or present since menarche and >12 months	Acute menorrhagia requiring hospital admission and emergency treatment or BT/RT/DDAVP or D&C or endometrial ablation or hysterectomy
Post-partum hemorrhage	No/Trivial or no deliveries	Consultation only ^a^ or use of syntocin or lochia > 6 weeks	Iron therapy or antifibrinolytic	BT/RT/DDAVP or requiring examination under anesthesia and/or the use of uterine balloon package to tamponade the uterus	Any surgical procedure requiring critical care or surgical intervention (e.g., hysterectomy, iliac artery ligation, uterine artery embolization, uterine brace sutures)
Muscle hematomas	No/Trivial	Post trauma, no therapy	Spontaneous, no therapy	Spontaneous or traumatic requiring DDAVP or RT	Spontaneous or traumatic requiring surgical intervention or BT
Hemarthrosis	No/Trivial	Post trauma, no therapy	Spontaneous, no therapy	Spontaneous or traumatic requiring DDAVP or RT	Spontaneous or traumatic requiring surgical intervention or BT
CNS bleeding	Never	-	-	Subdural, any intervention	Intracerebral, any intervention
Other bleeding	No/Trivial	Present	Consultation only ^a^	Surgical hemostasis or antifibrinolytic	BT/RT/DDAVP

Abbreviations: BT/RT/DDAVP: blood transfusion, replacement agents, desmopressin; CNS: central nervous system; D&C: dilatation and curettage; GI: gastrointestinal; PBAC: pictorial bleeding assessment chart; consultation only ^a^: the patient sought medical evaluation and was either referred to a specialist or offered a detailed laboratory investigation.

#### 2.1.2. The Patients with Thrombocytopenia

A platelet count less than 100 × 10^9^/L generally suggests the presence of a platelet disorder (primary or secondary) requiring further investigation [4,19], as do higher platelet counts with additional symptoms that may suggest a specific IPD. IT are an heterogenous group of disorders in terms of severity and molecular etiology, with a prevalence estimated to be around 2.7 per 100,000 people.

The focus of the clinical evaluation of a patient presenting with thrombocytopenia should include:
▪Spectrum of clinical manifestations: -Thrombocytopenia of variable severity and clinically significant bleeding (see above).-Thrombocytopenia, usually mild–moderate, asymptomatic, or associated with minimal/absent bleeding-Syndromic thrombocytopenia-Thrombocytopenia with predisposition or risk to other hematologic or extra-hematologic complications▪Platelet count and platelet morphology:


It is common to have some laboratory testing results available at the time of the history taking. It is essential to establish the onset or the age at which the thrombocytopenia was first recognized, its stability over time (absence of normal levels ≥ 175–200 × 10^9^/L), and the presence or absence of a family history of thrombocytopenia. This information may support the suspicion of an IT. However, in the absence of other information, isolated thrombocytopenia of an unexplained cause may be sufficient reason to suspect an IT.

Platelet size and morphology are important criteria for guiding the diagnosis, as they are characteristic of the different IT and prior complete blood counts and/or smears should be reviewed.

▪Associated systemic diseases (syndromes or predisposition):

It is well known that certain IT, as well as some IPFDs without thrombocytopenia, may present with other systemic manifestations, predispose to them, or be part of a complex syndrome [1,2,4,6,19] (Figure 2) (Table 1). These conditions may or may not be present at the time of taking the patient’s medical history. A careful review of the family history, in which these conditions may already be evident, and previous information provided by other specialists, as well as an adequate physical examination and/or appropriate complementary examinations of the complex syndrome, is recommended [1,2,4,6,19] (Figure 1 and Figure 2; Table 2).

#### 2.1.3. Patients with Other Atypical Presentations

Other patients in whom it is reasonable to suspect an IPD include:▪Patients with a previous diagnosis of ITP who do not respond to standard treatments, especially pediatric or young adult patients, and/or cases defined as familial ITP. It should be noted that, historically, a significant proportion of patients with ITs were/are erroneously diagnosed and treated with ITP, due to the higher incidence of this disease and the fact that general practitioners and hematologists are more familiar with this acquired pathology [5,20].▪Patients presenting with hematologic neoplasm without other predisposing factors may previously have had long-standing, unrecognized thrombocytopenia. Up to 30% of *RUNX1* disease-causing genetic variants in adult patients with acute myeloblastic leukemia have been reported to be germline [21]. Hence, patients could have previously been diagnosed with *RUNX1*-related thrombocytopenia (RUNX1-RT or *RUNX1*-FPD, or FPD/AML, or FPDMM).▪People with disproportionate thrombocytopenia in the context of pregnancy or “familial” heavy menstrual bleeding.

### 2.2. Family History of Patients with Suspected IPD

Obtaining a thorough family history and documenting the family’s pedigree is also important [1,6,10,22,23]. Some key aspects to consider are:▪The presence of relatives with clinical characteristics like those of the index patient, which increases the likelihood of them having an IPD.▪If affected relatives belong to several generations, an autosomal dominant (AD) IPD may be suspected.▪If there are no affected relatives, or the few that are belong to different generations, an autosomal recessive (AR) IPD is more likely. This also applies to families with known consanguinity.▪When the disease predominantly affects males, one can speculate an X-linked IPD inheritance [24,25].▪The absence of other affected family members should not rule out the existence of an IPD considering that:▪In cases of AR inherited IPDs in which only homozygous or compound heterozygous carriers manifest the disease, heterozygous subjects are usually asymptomatic.
-AD IPDs arising de novo in the patient, as occurs in 20–25% of patients with *MYH9*-related disorder (MYH9-RD) [1,24].-Patients with IPD show high variability in their clinical presentation, due to variable penetrance and the presence of other elements (congenital or acquired, known or unknown) that modulate clinical and laboratory phenotypes.

### 2.3. Other General Recommendations

In cases of adoption, estranged familial relationships, or in vitro *fertilization*, the complete family history may to be readily available.

Even in patients who do not report bleeding symptoms, an evaluation using standardized bleeding scores such as the ISTH-BAT is recommended.

Physical examination should focus in the evaluation signs of active bleeding, scars from previous trauma or surgery, or hyperlaxity of the skin and joints suggestive of collagen disease, as well as to try to rule out unrecognized diseases or syndromes potentially associated with bleeding [26].

## 3. Laboratory Studies

To assess the laboratory phenotype of a patient with suspected IPD we suggest, in line with the recommendations of other experts, sequential/staggered studies be obtained. These are typically performed in groups according to complexity and availability, with the most complex assays being performed in reference centers [1,7,8,9,10,23,27].

Considering the wide spectrum of platelet counts and bleeding in patients with IPDs, a common initial approach to laboratory testing is recommended for all types of IPDs. However, it should be noted that moderate to severe thrombocytopenia may prevent the performance of some more traditional and widespread platelet function tests or may require their modification and validation in samples with low platelet counts.

Figure 3 presents a phased laboratory study. According to the availability of assays and experience of the laboratory that will be performing the testing, variations in the sequence of studies is expected. In some cases, overlapping the phases of testing is necessary to avoid the need to schedule multiple appointments and minimize blood draws.

### 3.1. Phase 1: Initial Laboratory Studies

This first phase of testing aims to rule out VWD or other coagulation factor deficiencies, as well as assess the number and morphological characteristics of platelets and other blood cells. Essential tests to be performed include:(a)Coagulation studies:-Prothrombin time (PT);-Activated partial thromboplastin time (APTT);-FVIII activity;-Fibrinogen activity.


*Recommendations:*
○Prolonged screening of clotting times should prompt additional evaluation to identify specific coagulation factor deficiencies.
(b)Study of von Willebrand Factor (VWF):-VWF antigen level;-Functional activity of VWF.



*Recommendations:*
○If the results are abnormal and not due to pre-analytical errors, diagnostic algorithms should be followed to classify VWD by quantitative or functional VWF deficiency [28,29].○When interpreting the results, it is essential to take into account the ABO blood type, since individuals with type O usually have lower levels of VWF compared to other blood types [29].○APTT may be prolonged if plasma FVIII levels are sufficiently low, especially in VWD types 2N and 3.○In some cases, a coagulation disorder may coexist with platelet dysfunction. Therefore, platelet function studies may be recommended in patients with mild VWF deficiency or in those with known coagulation factor disorders who present with bleeding manifestations that are disproportionate or unusual to those expected in these conditions [30,31].
(c)Complete blood count (CBC):


The CBC is one of the first and most important tests in the evaluation of a patient with a suspected IPD. It is standardly performed on a hematology counter using EDTA blood. These devices are available in most hospitals, and although there are various modalities that employ different technologies, all provide the essential blood parameters.


*Recommendations:*


All the data from the complete blood count may be relevant in the context of the diagnosis of an IPD, in particular:○Platelet count and platelet size:

In general, the normal platelet count is estimated to be between 150–400 × 10^9^/L, and the mean platelet volume (MPV) between 7/8 and 11/12 fL. Normal, small, or large platelet size is a typical feature of the different types of HT, but it should be noted that there may be exceptions.

A cutoff of 12.4 fL has been reported to have a sensitivity of 83% and a specificity of 89% for distinguishing inherited macrothrombocytopenia from ITP [32]. However, it should be noted that when platelets are very large or giant, the counters may underestimate the platelet count and not provide a true MPV value. In these cases, reviewing the histogram provided by the automatic counter may be useful in detecting cases of macrothrombocytopenia. The histogram normally shows platelets as small cells with internal granules, approximately one-sixth to one-third the size of red blood cells, presenting a typical “bell-shaped” distribution. However, in cases of macrothrombocytopenia, platelets approach the size of a red blood cell and the histogram is distorted and shows a shift to the right [33].

○Red blood cell and leukocyte count:

The association of thrombocytopenia and red cell abnormalities is characteristic of certain IT subtypes, such as that associated with *GATA-1* disease-causing variants (GATA1-RT). The association of anemia or leukopenia may indicate underlying secondary causes that should be ruled out, such as hematologic malignancy or bone marrow failure. However, these associations can also be present in other types of IPDs.

○Other parameters of interest:

Some counters also offer the percentage of reticulated platelets (RP) or immature platelet fraction (IPF). Conceptually, these are young platelets and equivalent to reticulocytes in the red blood cell, so a higher-than-normal reticulated platelet count (>10–15%) can be used to distinguish macro-IT from thrombocytopenia caused by bone marrow failure. Although IPDs tend to have a higher IPF than ITP, the sensitivity and specificity ranges are not well established [34,35,36].

(d)Conventional peripheral blood (PB) smear:

This test is performed by smearing a few drops of blood in EDTA on a glass slide, staining with May–Grünwald–Giemsa (MGG), and observing the smear under a light microscope. Currently, there are devices that allow for the scanning of these PB smears and their automated analysis with specific software.


*Recommendations:*
○Platelet clumping and satellitism should be evaluated as a cause of pseudothrombocytopenia, an artifactual thrombocytopenia, usually in blood anticoagulated with EDTA, which occurs in up to 0.3% of cases [37].○Platelet aggregates can also occur in platelet-type VWD (PT-VWD), and its phenocopy VWD-type 2B.○Identification possible morphological and granular abnormalities in platelets:▪Platelet size determined with the mean platelet diameter (MPD) can guide the diagnostic process of a IT, as it allows for the identification of platelets that are considered large (MPD > 3.2 μm), normal (MPD 2.6–3.2 μm) or small (MPD < 2.6 μm) [38]. Microthrombocytopenia can be related to molecular alterations in the *FYB*, *ARCP1B*, *PTPRJ* (*CD148*), or *WAS* genes. In contrast, macrothrombocytopenia, even with giant platelets, is typical of alterations in the *GP1BA*, *GP1B*, *GP9*, *MYH9*, *SRC*, and *SLFN14* genes, and in the majority of ITs from variants in genes that encode cytoskeletal proteins or participate in glycosylation [1,2,4].▪The presence of poorly stained platelets in the smear is generally suggestive of a defect in the number or content of alpha granules. Gray or pale or highly vacuolated platelets associated with an α-granule deficiency caused by molecular abnormalities in genes such as *NBEAL2*, *GATA-1*, or *GFI1B*, among others, can be identified [1,2,4]. δ granules are not visible in MGG staining and must be evaluated by other methods such as “whole mount” electron microscopy (see below).○Morphologic abnormalities in other cell lines:▪Erythrocytes: Morphological abnormalities of red blood cells occur in cases of GATA1-RT (anisocytosis and poikilocytes), Stormorken syndrome (Howell-Jolly bodies caused by asplenia), or sitosterolemia due to variants in the *ABCG5/ABCG8 genes* of sterol metabolism (stomatocytes and possible signs of hemolysis) [1,2,4]. The presence of schistocytes together with thrombocytopenia, hemolysis, and renal failure suggests the diagnosis of thrombotic microangiopathy, an entity with which the differential diagnosis should be made.▪Leukocytes: It is essential to rule out the presence of dysplastic features or the presence of immature elements that suggest a possible underlying myeloid neoplasm. In some, but not all, patients with MYH9-RD, blue-stained cytoplasmic inclusion bodies, commonly called Döhle-like bodies, are observed in neutrophils. Immunofluorescent staining with antibodies against MYH9 is a much more sensitive alternative for detecting these MYH9 aggregates in neutrophils that are present in virtually all cases of MYH9-RD [8,39]. The presence of giant granules in leukocytes suggests the diagnosis of Chediak–Higashi syndrome (CHS).


Figure 4 shows examples of characteristic findings of specific disorders under conventional microscopy of the PB smear.

(e)Bone marrow study:

At present, there is no consensus as to recommendations for bone marrow aspiration/biopsy in patients with IT. Recommendations for bone marrow aspiration/biopsy are extrapolated from clinical practice guidelines for patients with other causes of thrombocytopenia and vary depending on smear findings, the appearance of unexplained cytopenias or immature cells that may suggest a possible hematologic malignancy [40].


*Recommendations:*
○In patients with IPD and high risk of bone marrow failure (CAMT, thrombocytopenia with radioulnar synostosis, or thrombocytopenia with absence of radius), bone marrow biopsy is recommended.○In patients with an IPD with a risk of developing hematological malignancies, a baseline bone marrow biopsy can be considered. If there are significant clinical changes such as splenomegaly, or relevant alterations in the complete blood count/peripheral blood smear (cytopenias or in cases of bone marrow fibrosis), a bone marrow biopsy is recommended.


### 3.2. Phase 2: Global Hemostasis and/or Platelet Function Assays

There are various devices on the market that perform automated or semi-automated tests on blood samples to assess global hemostasis, primary hemostasis, and/or platelet function. Their major advantage is that they almost eliminate operator-dependent variability and subjectivity in result evaluation. However, variability persists due to pre-analytical factors (extraction quality, pre-test sample handling/preservation, etc.). The different devices employ different technologies; many of the tests are performed under high-flow conditions and/or in the presence of coagulation activators or platelet agonists, attempting to simulate the physiological conditions of hemostatic thrombus formation.

The most well-known and used equipment is the PFA-100 (or 200), different thromboelastographs, Impact-R, Verify Now and the T-TAS, which is one of the most recently developed [41,42,43,44].

This review is not intended to detail these technologies. We simply emphasize that widely used assays, such as the PFA-100, have not demonstrated sufficient sensitivity and specificity in the diagnosis of IPD [10,41,45]. There still scarce experience with other devices like T-TAS [43,46].

When using these systems, it is important to consider that they may require specific characteristics in the blood samples to be analyzed, for example, extraction with specific anticoagulants that are not commonly used (BAPA or others), and a platelet and hematocrit count above a certain value.


*Recommendations:*
○Semi-automated global hemostasis tests, such as PFA-100 tests, are only optional in the diagnosis of IPD.○They can be useful to quickly exclude severe platelet dysfunction. For example, in the PFA-100 test, the closure times with both cartridges (Col-ADP and Col-EPI) are always abnormal (≥300 s) in cases of BSS or GT.○A normal or minimally abnormal result does not exclude mild to moderate platelet dysfunction. If the suspicion is high enough by the patient’s clinical symptoms, further testing is recommended.


### 3.3. Phase 3: Basic Studies of the Platelet Phenotype

(a)Platelet aggregation

Light transmission aggregometry (LTA), developed by Born and O’Brien in the early 1960s, remains the gold standard and most widely used method for studying platelet function [7,11,47,48,49,50]. The assay is based on measuring the increase in light transmission through a sample of platelet-rich plasma (PRP) kept under agitation, after inducing aggregation with a platelet agonist. The patient’s PRP is the most opaque sample (0% light transmission or 0% aggregation) and their platelet-poor plasma (PPP) the maximum achievable transparency (100% aggregation). The most classic aggregation pattern is biphasic, with a first wave corresponding to the primary response to the added agonist, and a second wave involving agonists produced and/or secreted from the granules in situ by the platelets themselves, the most relevant being ADP and thromboxane A2 (TxA2). The second phase, very typical in the case of epinephrine-induced aggregation, can be masked when high concentrations of agonists that induce intense primary responses are used. In addition to the amplitude of aggregation (% of maximum aggregation), other parameters such as the onset time, or lag time, and the slope or speed of aggregation are also characteristic of the platelet response to each agonist and can be analyzed to reveal specific functional alterations.

Despite being the gold standard, platelet aggregation has its limitations, including the fact that it requires a large sample volume, and a sufficient platelet count to obtain a valid PRP (>75 × 10^9^/L, depending on the equipment used). This may limit its use in children and patients with moderate-to-severe thrombocytopenia. Furthermore, it is a relatively expensive, laborious, and complex procedure that requires trained personnel and is influenced by many pre-analytical variables. To promote the standardization of LTA between laboratories, the ISTH Platelet Physiology Subcommittee developed a guideline of recommendations for aggregation assays [48], which is currently being revised, and other scientific societies and expert groups have provided recommendations for the performance and interpretation of LTA [10,47,49,50].

There are several models of aggregometers in the market (TA 4-V3 or TA 8-V3 from Stago Inc., Asnières-sur-Seine, France, Model 490 4 + 4 Optical from Chrono-Log, Havertown, PA, USA, or PAP-8E from Bio/Data Co., Horsham, PA, USA, etc.). Some of these devices are lumiaggregometers (such as the M700 from Chrono-Log) that measure light transmission and luminescence simultaneously. These lumiaggregometers use the luciferin–luciferase reaction principle to simultaneously analyze platelet aggregation and nucleotide (ATP) secretion. However, their utility in the diagnosis of IPDs is controversial, since the secretion results with individual agonists show a limited reproducibility [10].

On the other hand, platelet aggregation can also be measured in whole blood (WBA). The most traditional method uses impedance technology and is based on measuring the change in electrical resistance or impedance between two electrodes as platelets, activated by an agonist, adhere and aggregate to them. Several models of WBA instruments are commercially available, both semi-automated and with disposable cuvettes/electrodes pre-prepared for different agonists. The best known and most widely used is the Multiplate analyzer (Roche Diagnostic, Aperia, Singapore). WBA has the advantage of using a more physiological and unmanipulated sample, requiring less blood volume, and being faster. However, this method does not show the two aggregation waves compared to PRP, and the method is not exempt of the influence of the same pre-analytical variables that affect LTA. Furthermore, while WBA successfully identifies severe platelet dysfunction, it is considered less sensitive than LTA in detecting patients with moderate platelet dysfunction [10].


*Recommendations:*
○The general recommendations for LTA established by ISTH [48] (currently under review) should be followed.○A healthy control should always be analyzed in parallel with the patient’s sample (using identical sample extraction and handling).○It is not necessary for the patients to fast before testing, but they must not have consumed meals with very high fat content (highly lipemic plasma) within 12 h of sample collection.○Blood should be drawn from patients and controls, preferably in the morning, after a short rest (15–30 min), without having smoked (30 min), and without having taken supplements or medications that potentially inhibit platelets or NSAIDs (3 days), or irreversible antiplatelet agents such as acetylsalicylic acid (10 days).○Blood should be drawn with minimal compression and a 21 g needle; first, a sample in EDTA (2–4 mL) for complete blood count or other purposes, and then blood for LTA and other functional studies (usually 20–30 mL) in citrate tubes (9:1), preferably 3.2% (109 mM) and buffered, although 3.7% citrate, 129 mM is acceptable.○If commercial vacuum tubes with a fixed blood volume are used (the ones commonly found in hospitals), it is ideal to use those with a larger volume (10 or 5 mL, preferably over tubes of <3 mL), ensure that they are filled correctly, and that the blood mixes well with the anticoagulant.○Samples should be stored at room temperature (do not store in refrigerators or in coolers).○Avoid vigorously shaking the tubes. If mixing is necessary, do so by gentle inversion or using a tube rotator at low speed.○If samples must be transported to another laboratory or center, they must be transported in suitable containers with the tubes properly protected. This is especially important if the transport is long-distance and involves courier companies (instruct them not to refrigerate and handle carefully).○Complete the LTA study as soon as possible after blood collection. Guidelines recommend < 4 h [10,48], but we acknowledge this may be difficult to achieve.○The LTA study on 12 to 24 h blood samples sent by courier may be considered if there is no other option; for example, if the hospital does not have the methodology and/or the patient cannot travel to another center. In this case, it is essential to analyze samples from a healthy control collected and shipped in parallel, as well as samples from another healthy control collected just before the LTA test is performed (as a normal response control).○In our experience, in blood aged ≈ 24 h, there is an almost complete loss of aggregation with weak agonists such as epinephrine (any dose) or with low doses of ADP, TRAP, or others; with medium/high doses of these agonists, a moderate reduction in aggregation is observed (≈25–30%, sometimes the loss of the second wave); and with potent agonists such as CRP, arachidonic acid, or ristocetin, LTA is usually only slightly altered.○PRP is generally obtained by centrifuging the original tubes (usually 4–5 mL) at room temperature for 10 min at 150–200× *g*, with minimal or no brake. Transferring to other tubes, checking for lipemia, icterus, or hemolysis, and taking a sample for a complete blood count. The platelet count should be higher than that of the original blood.○In patients with large or giant platelets (such as those with various macrothrombocytopenias or BSS), standard centrifugation may result in the loss of a significant number of platelets, particularly large platelets. In these cases, it is possible to obtain PRP at a slower speed (below 100× *g*) or by gravity sedimentation after allowing the tubes to rest (≈1 h) at an inclination angle of 45 degrees. It should be noted that these PRPs may be more contaminated with leukocytes and red blood cells; this contamination can be reduced by allowing the PRP to rest again or by centrifuging at low speed (50× *g*, 5 min).○After separating the PRP, the platelet-poor plasma (PPP) obtained will be used as a blank by centrifuging the tubes at room temperature for 15 min at 1200–1500× *g*.○The recommended platelet count in PRP is 150–400 × 10^9^/L. The ISTH guidelines recommend not adjusting the platelet count of the PRP obtained, but if adjusted, using a suitable buffer [48] such as Tyrode’s buffer. However, many experts prefer to adjust the PRP count with PPP to 200–250 × 10^9^/L. ^10^ Our practice is to not adjust the PRP if it is in the range of 150–400 × 10^9^/L, but to adjust it with PPP to 250–300 × 10^9^/L if the count is >400 × 10^9^/L.○In IT patients, PRP may be lower than 150 × 10^9^/L; hence, adjusting the control PRP to the same level is recommended. Current aggregometers, such as the Stago TA 4-V3, can reproduce platelet aggregation with PRP levels of approximately 50 × 10^9^/L.○The LTA assay is performed at 37 °C. The equipment is thermostatically controlled. Recording is typically carried out for 5 min; for some uncommonly used agonists like PMA, it is ideal to extend the time to 10 min, as the aggregation response is slow. It may also take 10–15 min to observe disaggregation with agonists like epinephrine.○Ideally, each laboratory should have its own normal LTA range for both normal-count PRP (150–400 × 10^9^/L) and thrombocytopenic PRP (≈50–100 × 10^9^/L). Where appropriate, a normal range should also be available for PRP from blood samples from ≈24 h apart. The recommended number of controls for establishing these normal ranges is >40.○Table 3 shows the panel of agonists that we recommended in the first LTA study.○If the volume of PRP obtained is insufficient for the proposed panel, a more basic study is recommended with 1.5 mM arachidonic acid, 5 μM ADP, 25 μM TRAP, 1.25 mg/mL ristocetin, and 2 μg/mL collagen. This study may be sufficient to guide an IPFD in a first LTA study.○In each LTA study, it is advisable to repeat the test with any agonist if it gives an abnormal result, both in the patient and in the control.○Depending on the results of LTA obtained, consideration should be given to testing other doses and/or other agonists (PAR-4, A23187, PMA, convulxin, PAF, or others) in the same study, or in a subsequent study according to the diagnostic suspicion.○If a patient’s LTA study is abnormal, it should be repeated after at least one month to confirm the persistence of the abnormality.○In the global interpretation of the aggregation curves, the following must be assessed:▪Change in platelet shape: This is reflected by a reduction in light transmission at the beginning (10–30 s) of the aggregation curve. This is not evident with all agonists.▪Delay in the onset of aggregation (lag): with some agonists such as collagen, a lag of ≈30–50 s is typical.▪Aggregation slope: usually directly proportional to maximum aggregation. It is usually greater with strong agonists.▪Second wave of aggregation: Typical of epinephrine and weak agonists/low doses. Not evident with strong agonists/high doses.▪Maximum aggregation (%).▪Disaggregation: Reflects the formation of loose aggregates. If it occurs, final aggregation is less than maximum. It usually occurs with weak/low-dose agonists (ADP, TRAP, U46619) in moderate IPFD signaling/secretion, or due to platelet desensitization, for example, in PRP from non-fresh blood.○Some aggregation patterns in response to the recommended agonists are characteristic of specific types of IPFD and allow for their diagnosis. Others are indicative of a category of defects but are not specific and do not allow for the diagnosis of a specific type. There are multiple expert guides and publications with recommendations for interpreting aggregation patterns (Table 4) [10,51].


**Table 3 biomolecules-15-00846-t003:** Panel of first-tier agonists recommended for the study of in vitro platelet aggregation in patients with a suspected IPFD [52].

Agonist	Concentration	
	PRP Fresh Blood	PRP Blood 12–24 h *
Arachidonic Ac.	1.5 mM	1.5 mM
ADP If altered at low dose	2.5 μM 10 μM	5 μM 10 μM
Epinephrine If altered at low dose	5 μM 10 μM	-
TRAP (PAR-1) If altered at low dose	12.5 μM 25 μM	25 μM
Ristocetin If platelet type VWD is suspected	1.25 mg/mL 0.5–0.8 mg/mL	1.25 mg/mL 0.8 mg/mL
Collagen If altered at low dose	2 μg/mL 5 μg/mL	5 μg/mL 10 μg/mL
CRP (If altered with collagen) If altered at low dose	2 μg/mL 5 μg/mL	5 μg/mL 10 μg/mL
U46616 (If altered with arachidonic acid) If altered at low dose	2 μM 5 μM	5 μM 10 μM

* Only assess if unable to obtain fresh whole-blood sample. PRP: platelet-rich plasma.

**Table 4 biomolecules-15-00846-t004:** Hallmark aggregation patterns in different types of IPD [1,2,4].

TPC	LTA Pattern
BSS (GPIb/IX defect)	Aggregation absent/severely decreased with ristocetin; normal or reduced, but not absent, with other agonists (arachidonic acid, ADP, collagen, TRAP).
GT (defect of αIIbβ3)	Aggregation absent/severely decreased with all agonists (arachidonic acid, ADP, collagen, TRAP), except ristocetin (reduced but not absent). This pattern, perhaps less severe, can be seen in cases of dominant thrombocytopenia due to mutations in the Glanzmann genes (*ITGA2B* and *ITGB3*).
RASGRP2 defect (CalDAG-GEF)	Aggregation absent/severely decreased with low-moderate doses of most agonists (ADP, epinephrine, collagen, TRAP, etc.), but normal or minimally affected with high doses of these agonists, arachidonic acid, ristocetin or PMA (PKC activator).
GPVI deficiency	Aggregation absent/severely decreased with CRP (even high doses) and low doses of collagen; normal/less affected with other agonists.
P2Y12 receptor deficiency	Severely decreased and reversible aggregation with ADP even at high doses (>10 μM). Aggregation is also reduced with low–moderate doses of other agonists such as collagen, but normal or minimally affected with high doses of epinephrine (5–10 μM). The P2Y12 defect can be confirmed by other assays such as PGE1-induced cAMP formation in the presence and absence of ADP or epinephrine.
TxA2 receptor deficiency	Aggregation absent/severely decreased with arachidonic acid and U46619. Variably reduced with moderate–low doses of other agonists such as collagen or TRAP.
TxA2 generation deficiency (COX1 or Tx-synthetase defects)	Aggregation absent/severely decreased with arachidonic acid, but normal or minimally affected with U46619. Variably reduced/reversible with weak/low dose agonists (epinephrine, ADP, TRAP, collagen), reflecting the lack of secondary potentiation by TxA2.
Defect of dense granules (δ) or their secretion.	Aggregation absent/severely decreased with low–moderate doses of multiple agonists, in particular collagen at low doses, due to a lack of feedback by secreted ADP. Conversely, normal or minimally affected aggregation with high doses of ADP (5–10 μM). If lumiaggregometry is performed, ATP secretion is absent/severely decreased even with strong agonists/high doses. The granule defect may be confirmed by secretion assays (serotonin, CD63, etc.) or electron microscopy (whole mount; see below).
Signaling defect via Gi (ADP [P2Y12] or epinephrine receptors)	Aggregation absent or severely decreased and/or reversible with high doses of epinephrine and ADP (10 μM). Variably reduced with moderate–low doses of other agonists such as collagen or TRAP; normal or minimally affected with arachidonic acid.
Signaling defect via Gq/G_12/13_ pathway (thrombin receptors, TxA2)	Aggregation absent or decreased and/or reversible with moderate–low doses of TRAP or U46616, normal or minimally affected with ADP, epinephrine, CRP, or PMA.
Defects of transcription factors such as RUNX1 or ETV6	Variably reduced aggregation with moderate–low doses of multiple agonists. Less severe pattern than in GT or RASGRP2 defect.

BSS: Bernard–Soulier syndrome, GT: Glanzmann thrombasthenia.

(b)Platelet structural and functional studies by flow cytometry

Flow cytometry (FC) is a versatile technique fully implemented in the first line of diagnosis, namely the monitoring and characterization of hematological disorders, including IPD [10,53,54,55,56]. The principle is based on labeling platelets (diluted whole blood, PRP, or washed platelets) with one or more fluorescent antibodies (or other ligands) directed against specific antigens located on the platelet membrane (constitutively or after inducing platelet activation) or even intraplatelet antigens if the platelets are permeabilized. Cell samples are passed through a flow chamber equipped with one or more focused laser beams that affect the cells and activate the fluorophores. The equipment allows the sample volume and/or the number of cells (total or of a certain type) captured to be established. The analysis of the light-scattering patterns caused by cells and the intensity of the emitted fluorescence, using appropriate software, is used to identify platelets, their size and complexity, and to estimate the number of antibodies adhering to the platelet antigens evaluated.

FC has the advantage of requiring minimal blood volumes and platelet counts, making it an ideal method in the clinical setting for analyzing samples from newborns or young children, and from patients with severe thrombocytopenia. Furthermore, antibody-labeled samples can be fixed and analyzed hours later, making it feasible to send them to a central laboratory with appropriate equipment and experience.

The main application of FC is the identification of platelet glycoprotein (GP) deficiencies, such as GT (GP IIb/IIIa deficiency) and BSS (GP Ib/IX deficiency). It is also a valuable test for identifying alpha and dense granule deficiencies. Furthermore, it is worth mentioning that new methodologies such as mass cytometry are being developed, which, among other advantages, allows for the simultaneous evaluation of multiple components of individual cells, such as surface and intracellular proteins and platelet activation markers. It also allows the use of barcodes to analyze results from single cells or multiple samples at the same time [10,55].


*Recommendations:*
○Follow the ISTH guidelines to the application of FC in the study and research of IPDs, which describes the consensus among experts regarding clinical settings in which it is useful, pre-analytical variables, the standardization of instruments and reagents, methods of performance, and the reporting of results and quality control [53].○The FC study of any IPD should include the evaluation of the main GPs. Based on scientific evidence [10,11,53,54,55] and our experience, we recommend:▪Performing the test in whole blood (EDTA or citrate) diluted 1/10 with a buffer (1/5 if thrombocytopenic with platelets < 25 × 10^9^/L).▪Identify the platelet population with a tracer antibody against a major GP (CD41*PE; anti-IIb). For example, in cases of GT (lacking GPIIb), an anti-CD42B*PE (anti-GPIb) could be used. Alternatively, the platelet population can be identified by its size and scatter pattern.▪Fixating the sample with 2% PFA upon completion of labeling, and dilution of the sample before analysis.▪If the analysis of the samples is not carried out in within 2 h, samples should be stored at 4 °C for up to 18–24 h, and re-mixed well (gentle vortex before analyzing them in the cytometer.▪Table 5 shows a recommended GP analysis panel and reference intervals for each in healthy controls.○The FC study of a patient with suspected IPFD, with or without associated thrombocytopenia, should also include activation testing and agonist-induced granule secretion if functional FC is available. Based on our experience and scientific evidence [10,11,53,54,55], we recommend:▪Performing the assay on diluted PRP (≈10–20 × 10^9^/L) with Tyrode buffer. The assay is feasible on diluted citrated whole blood (1/10, or 1/5 if thrombocytopenic with platelets < 25 × 10^9^/L). In a basic test, the following tubes are prepared:○Diluted PRP.○Tracer antibody to identify platelets (CD41*APC; or a CD42B*APC or if the patient has GT]; alternatively, platelets could be identified by the size and scatter pattern.○Active integrin αIIbβ3: fibrinogen (Fg)*FITC, PAC-1*FITC antibody, or an anti-Fg Ab *FITC (would attach to the fibrinogen that binds the integrin).○P-selectin release ((α granules), or anti-CD63 *PE antibody (δ granules).○Platelet agonist at the appropriate final concentration.▪Incubate (30 min at room temperature), fix (PFA 2%), dilute (1/10) the sample with buffer, and run it through the cytometer. Capturing at least 5000–10,000 platelets (CD41*APC+ events) is recommended, assessing the % platelets+ and the median fluorescence in FL1 (Fg*488, PAC-1*FITC or anti-Fg*FITC) and FL2 (CD62*PE or CD63*PE). Establish the corresponding compensations or adjustments in the different fluorescence channels, depending on the technical characteristics of the cytometer used.▪Similarly to the analysis of GP, if testing is not performed within 2 h, storing the samples at 4 °C for up to 18–24 h is recommended.▪Table 6 shows a recommended panel of agonists for performing FC activation and secretion assays, and the reference intervals of each in healthy controls.○Mepacrine uptake and its agonist-induced release can be included in the flow cytometric study of IPFD to evaluate δ granules [10,57]. Mepacrine is an acridine derivative that emits green fluorescence (FL1) and is taken up by the δ-granules.


(c)Electron microscopy—whole mount.

Electron microscopy is a simple and validated method to quantify the number of δ granules in platelets. It is very useful in the diagnosis of IPDs with severe granular defects such as Hermansky–Pudlak Syndromes (HPS) or Chediak–Higashi Syndrome (CHS).

The method consists of placing a drop of PRP (≈20 μL) on the matte face of a FFC 200 Ni 50/b × FORMVAR/CARBON FILM 200 MESH, NICKEL grid, incubating for 15–30 s, removing the drop by placing the edge against a piece of filter paper, and washing the grid three times with a drop of water. The grid is then allowed to dry and observed under an electron microscope. δ granules are generally distinguished as rounded, opaque, deep black dots due to their high calcium and pyropolyphosphate content. Due to variability in morphology, experience is required for their accurate identification and count [10,58,59].


*Recommendations:*
○Whole-mount grids should be prepared as soon as possible (ideally <24 h)○The number of δ granules should be counted to be about 25–50 platelets, using a magnification scale of 4000 to 10,000X. Normal platelets have about 2–5 δ granules (although the reference varies between performing laboratories) compared to 0–2 in subjects with severe defects (HPS, SCH), or even >100 in giant platelets.○Repeating the test at least twice is recommended in patients with moderate granule deficiency to verify the consistency of findings.


### 3.4. Phase 4: Expanded Platelet Phenotype Studies

(a)Analysis of Peripheral Blood Smear by Immunofluorescence

Greinacher’s group has developed a methodology for the immunofluorescent staining of conventional blood smears, which can be useful in confirming the diagnosis and characterizing IPD. The procedure involves preparing fixating and incubating the smears first with antibodies or other ligands against specific platelet proteins, and then with fluorescent secondary antibodies against the former. Finally, the immunofluorescent staining pattern is assessed under a fluorescence microscope [39].

The method, although not widely used, has the great advantage of requiring only a few drops of blood to perform conventional smears (possible in any hospital), and these can be sent by conventional mail to a reference center that has the necessary antibodies, microscopes, and experience. The procedure has been recently validated, demonstrating its validity in reliably identifying common IPDs such as MYH9-RD, BSS, monoallelic BSS, GT, TUBB1-RT, GFI1B-RT, FLNA-RT, storage pool defect (δ-SPD), and WAS-RD [60]. For other types of IPD, the immunofluorescence pattern of the smears is not so unique or specific to allow an unequivocal diagnosis, but it can show platelet alterations that guide a subsequent functional study and facilitate the interpretation of genetic testing results. In fact, in many cases, DNA analysis will only identify novel molecular variants, which are typically classified as variants of uncertain significance and hence their involvement in the patient’s disease cannot be confirmed or ruled out based only on molecular data. The information obtained from immunofluorescence assays on smears can aid in the interpretation of the pathogenicity of such molecular alterations. Recent publications provide examples of these immunofluorescent staining patterns in different types of IPDs [39,60], as well as the list of recommended antibodies [60].


*Recommendations:*
○Given the complexity of performing and interpreting immunofluorescence results, which requires significant expertise and the need for a broad array of antibodies and sophisticated equipment, we recommend this test in the last phase of the IPD diagnostic process. However, specialized centers with the necessary equipment and experience in this methodology may consider performing it in earlier phases.○Non-specialized centers can consider this test in phases 1–3 if their work is limited to preparing smears and sending them by conventional mail to a specialized center with experience in this methodology.○Using antibodies already validated for this procedure, as well as a well-standardized methodology for performing the smears, is recommended [60].
(b)Platelet ultrastructure by transmission electron microscopy (TEM)


TEM is the gold standard for identifying ultrastructural abnormalities in platelets. It assesses the α and δ granules in an open canalicular system [59,61] whose deficiencies are typical in certain IPDs such as SPG, HPS, or Jacobsen syndrome, among others. Table 7 summarizes a TEM procedure for assessing platelet ultrastructure.


*Recommendations:*
○Anticoagulated whole blood can be used for TEM; we recommend using PRP where the platelet concentration is higher and there is less contamination with other cells.○Expertise is required to assess platelet ultrastructure and document potential abnormalities. Central imaging services supporting research often have software tools useful for these tasks.
(c)Platelet Aggregation Studies and/or Flow Cytometry


Based on the results of the previous studies, the following may be considered at this stage:•Extended platelet aggregation studies with less conventional agonists (PAR-4, PAF, A23187, DTT, rhodocetin, CD9, combinations of agonists with inhibitors, or others).•Extended FC studies to detect other membrane receptors (P2Y1/P2Y12, Clec-2, GPIV, CD36, or others); calcium mobilization; intraplatelet proteins (VASP, others); procoagulant activity (Annexin-V binding); ristocetin-induced VWF binding; etc.•The methodology to be applied is like that described above, but it is advisable to have experience in the tests to be performed, hence why they are preferably performed in specialized centers.
(d)Clot retraction studies

Clot retraction is an essential process for hemostasis, tissue repair, and defense against infection. Essentially, this process occurs through the coordinated action of activated αIIb β3 receptors and the platelet contractile apparatus, which pull on the fibrin fibers in the thrombus. The clinical relevance of this process is evident from its deficiency in severe platelet diseases such as GT, characterized by significant bleeding. There are several methods for measuring clot retraction, ranging from assessing the clot volume/area before and after retraction to using advanced technologies to measure the individual forces generated by platelets when contracting [62,63,64].

(e)Quantification of second messengers, cytokines or other bioactive substances, granule proteins, membrane receptors or intraplatelet proteins

Based on the results of the prior studies, it may be reasonable to consider at this stage as well:•Obtaining and storing serum to quantify TxB2 (stable metabolite of TxA2) synthesis by ELISA or other alternative method. This compound is primarily synthesized by platelets and is essential for feedback on platelet reactivity to multiple agonists. Abnormal TxA2 synthesis may suggest functional/genetic alterations in enzymes such as cyclooxygenase and/or thromboxane synthetase.•Obtaining and preserving plasma for potential quantification (ELISA, HPLC, others) of bioactive substances or antiplatelet antibodies.•Obtaining and preserving supernatants from platelets stimulated with agonists, with or without simultaneous recording of aggregation, for the quantification (ELISA, HPLC, others) of second messengers (cAMP, cGMP, or others, in samples from lysed platelets by freeze–thaw or platelets, or using a platelet lysis agent), or proteins secreted from α granules (VWF, b-GT, PAF, Factor V, P-selectin, etc.) or δ granules (serotonin, ADP, ATP, etc.).•Preparing washed platelets and from them platelet lysates for the quantification of specific platelet proteins (membrane, cytosolic, granular), by electrophoresis and Western blot (WB) with specific antibodies•The methodologies used are diverse and can be found in numerous publications in journals and books on platelet research methods [65,66]. Experience in the assays is required; hence, these tests are preferably performed in specialized centers.
(f)Other research studies:

These include platelet-spreading assays on crystals coated with adhesive proteins (pre-adding on collagen, fibrinogen, or others); perfusion chamber studies; platelet proteome and/or transcriptome studies (arrays, RNAseq, etc.), among others.

The general recommendation is to review the literature related to the specific type of IPD suspected or diagnosed and seek expert advice.

### 3.5. Phase 5: Molecular Studies: Genetic Diagnosis

The definitive diagnosis of IPD is achieved by identifying the underlying molecular pathology. Traditionally, genetic testing of IPD was considered the final step in the diagnostic process and unnecessary in most cases [7,23]. This view has changed radically, and today, early and accurate molecular diagnosis of IPD is viewed as highly necessary and beneficial for the proper clinical management of patients with IPDs in general. It may even be considered essential in severe, life-threatening IPDs, in which there is a clear relationship between the genotype and the prognosis of the hematological and/or non-hematological disease, such as CAMT, MYH9-RD, WAS-RD, HPS, CHS, or RUNX1-RT.

The approach to the molecular study of IPD has evolved significantly in the last two decades, as a result of advances in sequencing technologies [67,68,69,70]. Classically, genetic testing was performed almost exclusively by the Sanger sequencing of candidate genes identified based on the clinical and laboratory phenotype of patients. This strategy, although highly useful, is laborious and costly, and is not applicable to patients with a nonspecific phenotype. Therefore, it is being abandoned as a first-line diagnostic tool and is becoming a technique for confirming molecular alterations and family studies.

Currently, the genetic diagnosis of IPD is primarily addressed using high-throughput sequencing technologies, known as HTS or NGS. Without going into methodological details, which are beyond the scope of this review, HTS is used for the simultaneous sequencing of:▪Pre-selected gene panels (variable between 10 and 300 genes).▪Whole exome (WES, coding portion [exons] and flanking regions of all genes; 1–2% of the genome).▪Whole genome (WGS, complete sequence of all genes).

Although different HTS methods have their advantages and limitations [67,68,69,71], these new methodologies have demonstrated diagnostic rates of 20% to 50%, depending on the population studied and the methodology used [72,73,74,75,76].

Cost reductions have allowed for greater access to HTS at the clinical and research level, and in many cases has come to the forefront of diagnostic approach to IPD. However, significant challenges remain in the genetic diagnosis, such as filtering and correctly interpreting the massive amount of molecular information, the limited curation and low reliability of available public molecular databases, the management of possible incidental findings, the complexity of variant interpretation which requires functional and/or familial data, and the relevant underlying ethical implications [69,70,77]. International standards and ethical protocols still need to be further developed and made accessible to safely integrate mass molecular testing and genetic diagnosis into clinical practice.

Structural molecular events such as large deletions, insertions, duplications, or inversions require complementary methods such as copy number analysis (CNV) (can be performed with HTS), aCGH (comparative genomic hybridization arrays), MLPA (multiple ligation-dependent probe amplification), and more recently long-read sequencing, also need to be used to obtain a comprehensive molecular evaluation [71,78].


*Recommendations:*


Considering the guidelines of scientific societies and experts and our experience, our recommendations for the genetic diagnosis of IPD are as follows:○If the suspected diagnosis of IT is well supported by the clinical phenotype, available laboratory data, and/or family history, molecular testing should always be considered from the beginning of the diagnostic process. This applies to all types of IT, even those that can be unequivocally diagnosed with other methods without the need for molecular testing (such as BSS and GT, among others).○It is essential to inform patients and their families of the objectives of molecular testing and its possible results and consequences, including the possibility of not reaching a definitive diagnosis due to technical limitations and the possibility of unexpected or difficult-to-interpret findings (uncertain significance). They should also be informed that in many cases, genetic diagnosis can aid in clinical management and allow for family screening and genetic counseling, but that it will not always provide a direct benefit in the treatment or progression of their disease.○It is essential to obtain informed consent (IC) from the patient before performing the molecular studies, including specific options for patients to decide whether they wish to receive information about incidental findings, variants associated with risk of malignancy or other serious diseases, including carrier status for recessive disorders.○When informing patients and obtaining the IC, applicable legislation (data protection, etc.) and the recommendations of scientific societies such as ISTH should be observed [77].○The type of methodology to be used and the most appropriate time to perform the molecular study depends on the type of suspected IPD, the importance of the molecular study both in the diagnosis of the disease (whether or not it can be diagnosed with other laboratory studies), as well as in its prognosis and clinical management, the accessibility of the methodology, the ability to cover its cost, and the experience of the group or laboratory genetic diagnosis in general.○Possible methodological approaches include:▪Familial screening for specific genetic variants: If the pathogenic variant is known in a patient, their relatives can be screened using Sanger sequencing, targeted Nanopore sequencing, or potentially other targeted methods.▪Single-gene Sanger sequencing: This may be a valid option for monogenic IPD with very specific clinical and phenotypic features. Examples include: BSS (*GP1BA*, *GP1BB*, *GP9 genes*); platelet-type von Willebrand disease (*GP1B1*); GT (ITGA2B and ITGB3); MYH9-RD if there is macrothrombocytopenia, Döhle bodies, and/or typical extra hematologic manifestations (*MYH9*). However, considering the current costs of HTS, and that the genes to be sequenced can be large and complex, our recommendation is to use HTS directly, allowing for the relatively rapid sequencing of multiple genes.▪HTS sequencing with gene panels: This is the most common, practical, and least expensive approach currently available. When selecting genes for these panels, we recommend following the recommendations of expert groups such as the ISTH [79] (https://www.isth.org/page/GinTh_GeneLists, accessed on 29 May 2025). A negative result, assuming the disorder is inherited, may be due to methodological limitations or to the fact that the causal gene is not included in the panel. In practice, some laboratories sequence the whole exome, but the analysis is restricted, as a first option, to panels of these selected genes.▪WES and WGS sequencing: This is appropriate for cases with a clear suspicion of IPD, where there is no candidate gene, and HTS gene panel testing is negative. WGS offers the additional advantage of exploring the non-coding DNA region, allowing for the detection of variants in distal and proximal regulatory elements. The chances of successful diagnosis with WES or WGS increase considerably if DNA is analyzed not only from the patient but also from other family members (affected and non-affected at minimum using a trio approach).▪Single-gene sequencing, WES or WGS by long-sequence sequencing (4th generation sequencing): currently reserved for cases with unequivocal suspicion of a specific IPD, in which conventional HTS sequencing (of short sequences) has been negative, perhaps due to its inability to detect complex structural variants [71,78].○The validation and interpretation of the molecular data obtained is a critical aspect for the success of genetic diagnosis and should always include the patient’s clinical and laboratory context. Therefore, either directly or through appropriate bioinformatics and/or genetics services, the following should be considered:▪Applying well-established sequencing quality criteria (depth, read quality, sequence alignment with the reference genome, etc.)▪Applying efficient variant filtering (eliminating common or low-frequency variants based on population databases such as gnomAD; adjusting frequency thresholds according to the dominant or recessive inheritance pattern; prioritizing according to the probable effect on proteins, etc.).▪Consulting databases of specific disorders and molecular variants such as ClinVar (https://www.ncbi.nlm.nih.gov/clinvar/, accessed on 29 May 2025), ClinGen (https://clinicalgenome.org/, accessed on 29 May 2025) or ISTH resources such as Gold Variant database [80], to establish associations of previously known variants with IPD phenotypes.▪Applying internationally recognized criteria, such as those in the ACMG/AMP guidelines, to classify variants as pathogenic, likely pathogenic, of uncertain significance, likely benign, or benign [81]. ClinGen includes expert panels that have adapted the ACMG/AMP rules for certain types of IT, such as GT [82] RUNX1-RT [83].▪Using web platforms such as Varsome (https://varsome.com/, accessed on 29 May 2025) that provide extensive information on genetic variants and propose a preliminary classification of their pathogenicity according to ACMG/AMP criteria.○Working as an interdisciplinary team in the final evaluation of the candidate variants to be reported, considering the experimental data, the possible co -segregation, and the strength of the genotype–phenotype relationship.○Avoiding reporting variants in genes without sufficient information to support a gene–disease relationship.○Reviewing previously identified VUS variants if new relevant evidence emerges.○Incorporating advances in technology and genetic discoveries to improve diagnosis.○Include patients with a confirmed genetic diagnosis of a IPD in a patient registry such as the Spanish Registry of Inherited Platelet Disorders (RETPLAC) (https://retplac.imib.es/, accessed on 29 May 2025).

Figure 5 graphically summarizes the approach to the genetic diagnosis of IPD.

## 4. Conclusions

The diagnosis of inherited platelet disorders remains a challenge due to their clinical and molecular heterogeneity, as well as the limited sensitivity of available tests. An integrated diagnostic approach, combining clinical history, functional studies, and genetic analysis, is essential for accurate identification. This not only prevents misdiagnosis and inappropriate treatment but also allows for appropriate genetic counseling and the optimization of patient management. The implementation of advanced methodologies and a multidisciplinary approach are key to improving the recognition and treatment of these disorders.

## Figures and Tables

**Figure 1 biomolecules-15-00846-f001:**
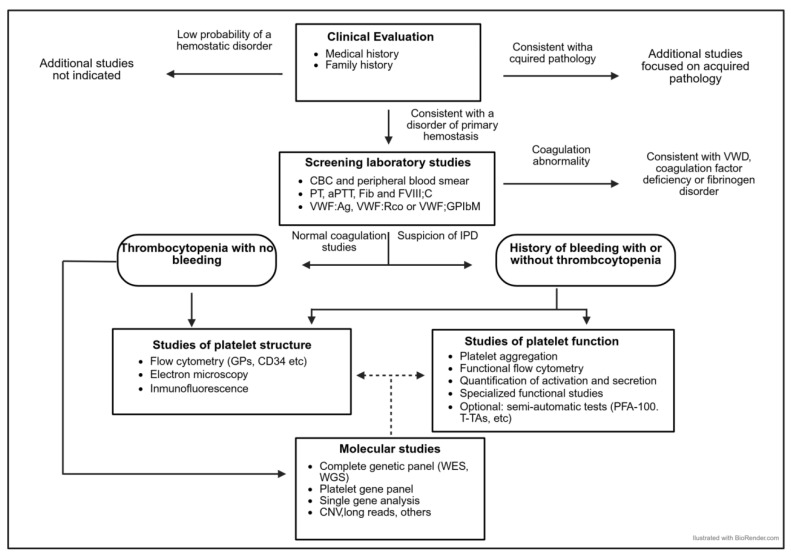
The diagnostic algorithm for patients with a suspected IPD.

**Figure 2 biomolecules-15-00846-f002:**
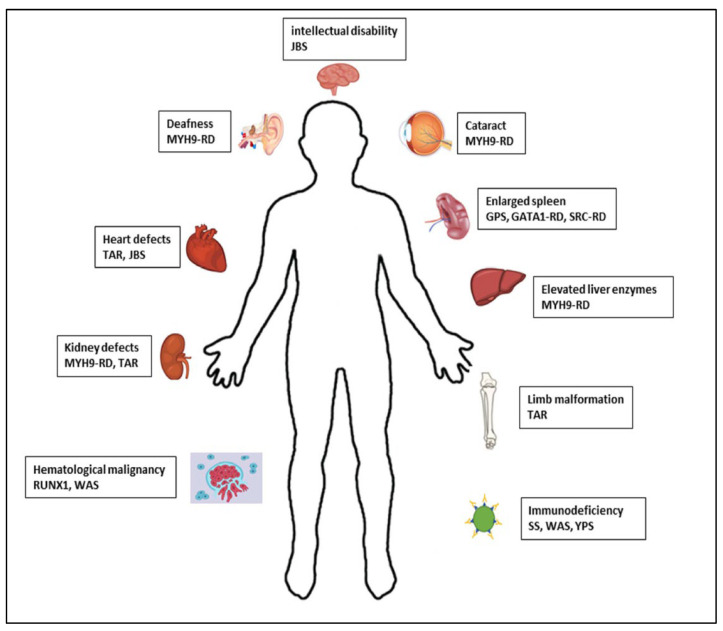
Syndromes and associated manifestations of IPD.

**Figure 3 biomolecules-15-00846-f003:**
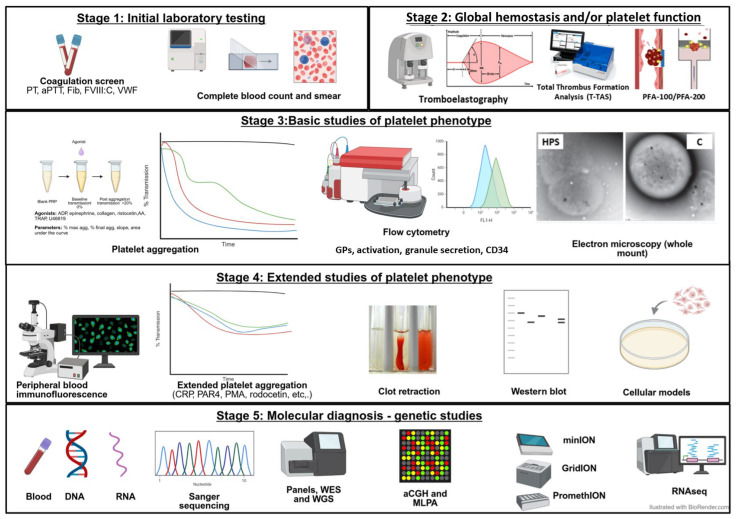
Staged laboratory evaluation of patients with IPDs.

**Figure 4 biomolecules-15-00846-f004:**
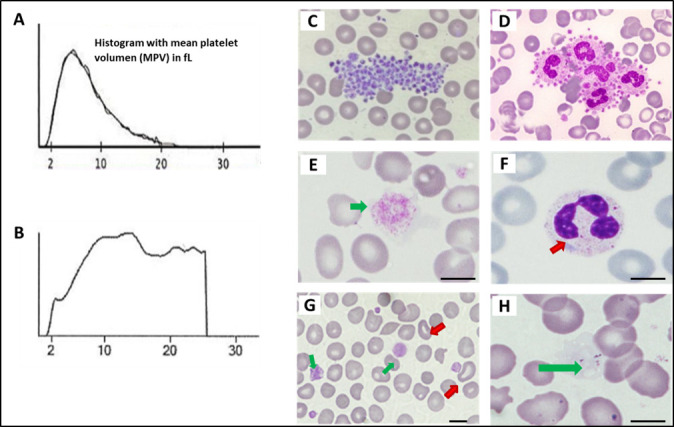
Platelet and other blood cell abnormalities on peripheral smear in patients with inherited platelet disorders. (A) Normal platelet histogram. (B) Abnormal platelet distribution in a patient with MYH9-RD with large/giant platelets. (C) Platelet clumping (image obtained from the Atlas of the Spanish Hematologic Cytology group [GECH]; https://atlas.gechem.org/index.php?lang=es, accessed on 29 May 2025). (D) Platelet satellitism (image from the GECH Atlas). (E) Giant platelet in a patient with BSS (GEAPC). (F) Döhle bodies in a neutrophil of a patient with MYH9-RD (GEAPC). (G) Stomatocytes and large platelets in a patient with Sitosterolemia (GEAPC). (H) Gray platelets due to absence of alpha granules in a patient with gray platelet syndrome (GEAPC). Scale bars are 5 μm.

**Figure 5 biomolecules-15-00846-f005:**
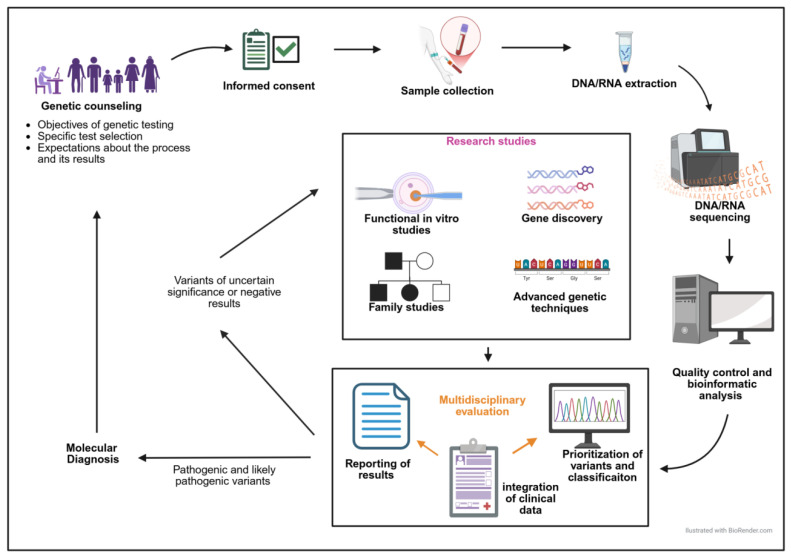
Genetic diagnosis workflow in patients with a suspected IPD.

**Table 2 biomolecules-15-00846-t002:** Clinical Associations, Syndromes and Related Disorders in IPD [1,2,4].

System	Clinical Manifestation	IPD (Genes)	Type of IPD	Platelet Size
Cutaneous	Eczema	Wiskott–Aldrich Syndrome (*WAS*)	IT	Small
	Xanthomas in tendons and xanthelasmas	Sitosterolemia (*ABCG5, ABCG8*)	IT	Large
	Oculo-cutaneous albinism	Hermansky–Pudlak Syndrome (*HPS*-1 to 11; 11 subtypes) Chediak–Higashi Syndrome (*LYST*)	IPFD	Normal
	Ichthyosis	Progressive variable erythroderma (*KDSR*) Stormorken Syndrome (*STIM1*)	IT IPFD	Large Normal
Sensory Organs	Hearing loss	DIAPH1-RT (*DIAPH1*) MYH9-RD (*MYH9*) MECOM-RD (*MECOM*)	IT	Large Large Normal
	Waterfalls	MYH9-RD (*MYH9*)	IT	Large
Immunological	Immunodeficiency	Leukocyte adhesion deficiency (LAD) type III (*FERMT3*) Chediak–Higashi Syndrome (*LYST*) Wiskott–Aldrich Syndrome (*WAS*)	IPFD IPFD IT	Normal Normal Small
	Hypogammaglobulinemia	Roifman Syndrome (RNU4ATAC)	IT	Normal
Hematological	Hemolytic anemia	Sitosterolemia (*ABCG5, ABCG8*)	IT	Large
	Erythrocytosis	ANKRD26-RT (*ANKRD26*)	IT	Normal
	Hematopoietic neoplasia	ANKRD26-RT (*ANKRD26*) ETV6-RT (*ETV6*) RUNX1-RT (*RUNX1*) ERG-RT (*ERG*)	IT	Normal
	Hemoglobinopathy	GATA1-RT (*GATA1*)	IT	Normal
	Bone marrow failure	MECOM-RD (*MECOM*) Congenital amegakaryocytic thrombocytopenia (CAMT) (*MPL*) THPO-RT (*THPO*)	IT	Normal
	Neutropenia	Hermansky–Pudlak Syndrome types 2 and 10 (*HPS2, HPS10*) Wiskott–Aldrich Syndrome (*WAS*) DIAPH1-RT (*DIAPH1*)	IPFD IT IT	Normal Small Large
	Myelofibrosis	G6b-B-RT (*MPIG6B*) Gray platelet syndrome (GPS) (*NBEAL2*) SRC-RT (*SRC*)	IT	Large
	Splenomegaly	Gray platelet syndrome (GPS) (*NBEAL2*)	IT	Large
Musculoskeletal	Radius and/or ulna malformation	Radioulnar–ulnar synostosis with thrombocytopenia (*RUSAT1*) (*HOXA11*) MECOM-RD (*MECOM*) Thrombocytopenia with absent radii (TAR) (*RBM8A*)	IT	Normal
	Myopathy	GNE-RT (*GNE*) Stormorken Syndrome (*STIM1*)	IT IPFD	Large Normal
	Facial dysmorphia	Baraitser–Winter Syndrome (BWCA) Takenouchi–Kosaki Syndrome (*CDC42*) Roifman Syndrome (*RNU4ATAC*)	IT IT IT	Large Large Normal
	Delayed growth	Roifman Syndrome (*RNU4ATAC*) Stormorken Syndrome (*STIM1*)	IT IPFD	Normal Normal
	Arthrogryposis	Arthrogryposis syndrome, renal dysfunction and cholestasis (*ARC*) 1 and 2 (*VIPAS39, VPS33B*)	IPFD	Normal
Cardiovascular	Atherosclerosis	Sitosterolemia (*ABCG5, ABCG8*)	IT	Large
	Structural congenital heart disease	MECOM-RT (*MECOM*) DiGeorge syndrome (22q11.2 deletion)	IT IT	Normal Large
Digestive	Cholestasis	Arthrogryposis syndrome, renal dysfunction and cholestasis (*ARC*) 1 and 2 (*VIPAS39, VPS33B*)	IPFD	Normal
	Granulomatous colitis	Hermansky–Pudlak syndrome types 1 and 4 (*HPS1, HPS4*)	IPFD	Normal
	Duodenal ulcers	Phospholipase A2c-RD (*PLA2G4A*)	IPFD	Normal
Respiratory	Pulmonary fibrosis	Hermansky–Pudlak Syndrome type 1, 2, 4 (*HPS1, HPS2, HPS4*)	IPFD	Normal
Neurological	Intellectual disability	Baraitser–Winter syndrome (*BWCA*), Takenouchi–Kosaki syndrome (*CDC42*) Roifman Syndrome (*RNU4ATAC*)	IT IT IT	Large Large Normal
	Periventricular heterotopia	FLNA-RT (*FLNA*)	IT	Large
Lymphatic	Lymphedema	Takenouchi–Kosaki Syndrome (*CDC42*)	IT	Large
Genitourinary	Congenital renal-urinary malformation	MECOM-RD Arthrogryposis syndrome, renal dysfunction and cholestasis (*ARC*) 1 and 2 (*VIPAS39, VPS33B*)	IT IPFD	Normal Normal
	Nephropathy	MYH9-RD	IT	Large

ANKRD26:Ankyrin Repeat Domain Containing 26; DIAPH1: Diaphanous-Related Formin 1; ERG: ETS Transcription Factor ERG; ETV6: ETS Variant Transcription Factor 6; FLNA: Filamin A; G6b-B: Megakaryocyte and platelet inhibitory receptor G6b; IPD: inherited platelet disorder; RT: related thrombocytopenia; RD: related disorder; IT: inherited thrombocytopenia; IPFD: inherited platelet function defect MYH9: myosin heavy chain 9; RUNX1: RUNX Family Transcription Factor 1; THPO: Thrombopoietin.

**Table 5 biomolecules-15-00846-t005:** A panel of monoclonal antibodies used for the analysis of platelet glycoprotein expression by flow cytometry and their reference interval in healthy subjects.

GPs	Antibody	Median Fluorescence of Healthy Controls in the Accuri C6 Cytometer (Mean ± SD)	*n*
Ib	CD42b*FITC	15,647 ± 3935	109
IX	CD42a*FITC	17,578 ± 6764	113
IIb	CD41a*PE	58,245 ± 15,374	113
IIIa	CD61*FITC	39,339 ± 7859	101
Ia	CD49b*FITC	1545 ± 845	113
VI	GPVI-APC	3877 ± 919	111
-	Nonspecific Ig-FITC	290 ± 172	88
CD34	CD34*APC	231 ± 58	40
-	FSC (size)	127,773 ± 29,148	121
-	SSC (granularity)	11,906 ± 2694	120

* All these antibodies from BD Biosciences (https://www.bdbiosciences.com/en-es, accessed on 29 May 2025). GPs: glycoproteins; FSC: forward scatter; FITC: Fluorescein isothiocyanate; APC: Allophycocyanin.

**Table 6 biomolecules-15-00846-t006:** Reference Intervals for Agonist-Induced Platelet Activation (Fibrinogen Binding) and Granule Secretion (CD62 and CD63 Expression) Assessed by Flow Cytometry in Healthy Individuals from the Author’s Laboratory.

Median Fluorescence in Healthy Controls (*n* = 74–116). Accuri C6 Cytometer			
Agonist	Fg*FITC αIIbβ3 active	CD62*PE (α granules)	CD63*PE (δ granules)
Tyrode Buffer	467 ± 147	273 ± 71	510 ± 123
ADP—10 μM	1494 ± 728	1395 ± 697	-
TRAP—25 μM	2891 ± 1779	4719 ± 2033	3268 ± 1433
PAR-4—1250 μM	5208 ± 3975	6321 ± 3237	-
CRP—10 μg/mL	9154 ± 6984	6726 ± 106	4059 ± 1341
U46619—5 μM + ADP—2,5 μM	8819 ± 6154	7263 ± 2775	-
PMA-100 nM	4823 ± 4179	4222 ± 1996	-

**Table 7 biomolecules-15-00846-t007:** Transmission electron microscopy procedure for evaluating platelet ultrastructure.

▪ Obtain anticoagulated whole blood; many protocols recommend ACD-A or B, but good results can be obtained using citrate or EDTA. PGE (100 ng/mL) or PGI2 can be added to avoid activation. ▪ Obtain the PRP by centrifugation 200× *g* 10 min and separate to another tube. It is important to obtain as much of the buffy-coat layer to not lose the larger sized platelets. Let the PRP sit at 37 °C for 15 min so that the platelets recover their discoid morphology well. ▪ In a 1:10 ratio, fix PPR 1.25% glutaraldehyde fixative solution in White A–White B (pre-warmed to room temperature) (10 mL round bottom tubes; mix by inversion), for 60 min at RT. ▪ Centrifuge at 1000× *g* for 5 min, remove the supernatant (aspiration or decanting), and gently resuspend the pellet in 10 mL of White A–White B wash solution. ▪ Repeat the previous washing step two more times. ▪ Resuspend the final platelet pellet in 2 mL of White A–White B wash solution. ▪ Take the samples to the central microcopy service for processing with OsO4, inclusion in EPON, and preparation of grids. ▪ Place the grids in the electron microscope, observe in different fields, and take a series of pictures (5–10) of each sample at different magnifications (2500X, 8000X, 13,500X). ▪ Expertise is required to assess platelet ultrastructure, and to document possible abnormalities. Central research support imaging services often have useful software tools available for these tasks. ▪ Reagents required: ▪ 1.25% glutaraldehyde fixative solution in White A–White B - 8.5 mL of water; - 0.5 mL of White A; - 0.5 mL of White B; - 0.55 mL of 25% glutaraldehyde; - Adjust pH to 7.4 (with 0.1 N ClH or 0.1 N NaOH). ▪ Washing solution - 27 mL of water; - 1.5 mL of White A; - 1.5 mL of White B; - Adjust pH to 7.4 (with 0.1 N ClH or 01 N NaOH).

## Data Availability

The original data presented in this study are available from the corresponding author upon reasonable request.

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
