# Peer review of "Diagnosis of Inherited Platelet Disorders: Clinical Evaluation and Functional and Molecular Assays"

_biomolecules, 2025, doi:10.3390/biom15060846_

Round 1

Reviewer 1 Report

Comments and Suggestions for Authors

Sanchez-Fuentes et al have completed a well written review on diagnosis of inherited platelet disorders. I have only minor comments

1) I feel some of the figures could do with being larger.  For example Figure 3 which is busy and the text is small as it stands.

2) The authors recommend repeating assays to confirm findings, what about running samples in duplicate or triplicate as well?  Should this be part of the usual way to run these assays?  There is no indication that it is.

3) Why would obtaining a supernatent from platelets help to identify cAMP or cGMP?  These are internal to the platelet so require lysis.

3) There are some minor editing issues.

Page 14 line 398, what do the authors mean by high and limited reproducibility?  Do they mean some agonists it is high and others it is not? 

Page 15 Line 461 I presume the authors mean 45 degrees as an angle rather than 45C?

Page 16 Table 3, the table seems to be misaligned with the CRP data. 

The authors should have a space between number and unit.  Table 3 is an example where this is not completed.

Table 6 the units are missing the micro sign I presume.

Page 23 line 749 I presume should read in many cases not it many cases.

Space required on line 754 after the references.

Author Response

REPLY TO REVIEWER 1

Comments and Suggestions for Authors

Sanchez-Fuentes et al have completed a well written review on diagnosis of inherited platelet disorders. I have only minor comments

We really thank the Reviewer for his/her very positive feedback on our manuscript.

1) I feel some of the figures could do with being larger.  For example Figure 3 which is busy and the text is small as it stands.

Thank you very much for the suggestion! I suppose the size of the figures will be determined by the magazine's printing requirements. In response to your suggestion, however, we have increased the size of the figures by 30% in this revised version, R1.

2) The authors recommend repeating assays to confirm findings, what about running samples in duplicate or triplicate as well?  Should this be part of the usual way to run these assays?  There is no indication that it is.

Thank you for your comment. We are not sure if you are referring to a specific test or in general. Whether we perform single, duplicate or triplicate assays depends on the complexity and reproducibility of the assay, and above all on the limitations of the patient's sample.

Rather than performing intra-assay duplicates or triplicates of everything, we recommend repeating a given test if an abnormal result is obtained to confirm it.  It is also highly recommended that the patient is retested after a period of time to confirm that they maintain a normal platelet function pattern.  For example, in our recommendations for platelet aggregation, we mention this on page 17:

"In each LTA study, it is advisable to repeat the test with any agonist if it gives an abnormal result, both in the patient and in the control."

"If a patient's LTA study is abnormal, it should be repeated after at least one month to confirm the persistence of the abnormality."

3) Why would obtaining a supernatent from platelets help to identify cAMP or cGMP?  These are internal to the platelet so require lysis.

Thank you for your comment, and we apologise for the misunderstanding. The reviewer is correct that cAMP and cGMP are second messengers generated and acting intraplatelet, and measuring them requires lysing platelets. We have indeed measured cAMP levels in platelets in previous studies, using platelets that were lysed by freeze-thaw cycles (see, for example, PMID: 15147378 and PMID: 31786577). To clarify this, we have modified the corresponding sentence on page 25 of the revised version of the manuscript to read:

“Obtaining and preserving supernatants from platelets stimulated with agonists, with or without simultaneous recording of aggregation, for quantification (ELISA, HPLC, others) of second messengers (cAMP, cGMP, or others, in samples from lysed platelets by freeze-thaw or platelets, or using a platelet lysis agent)….

3) There are some minor editing issues.

Page 14 line 398, what do the authors mean by high and limited reproducibility?  Do they mean some agonists it is high and others it is not? 

Thank you for your comment. The sentence was not phrased very well. We have corrected it in the new version, which now reads as follows: “..since the secretion results with individual agonists show a limited reproducibility10.

Page 15 Line 461 I presume the authors mean 45 degrees as an angle rather than 45C?

The reviewer is right. We have corrected the sentence, which now reads: “after allowing the tubes to rest (≈1 h) at an inclination angle of 45 degrees.”

Page 16 Table 3, the table seems to be misaligned with the CRP data

Thank you very much for bringing this to our attention. We have corrected Table 3 to state that the use of CRP is proposed if the response to collagen is altered. We have also included U46619, which was omitted by mistake.

The authors should have a space between number and unit.  Table 3 is an example where this is not completed.

This error has been corrected in the tables and in the text.

Table 6 the units are missing the micro sign I presume.

We have revised Table 6 and corrected minor errors

Page 23 line 749 I presume should read in many cases not it many cases.

The reviewer is right. We have corrected this..  “in many cases”

Space required on line 754 after the references.

Corrected

Reviewer 2 Report

Comments and Suggestions for Authors

In this Review, the Authors described the diagnostic algorithm to conduce to diagnosis of inherited platelet disorders.

My opinion regarding this article is positive. The authors comprehensively reviewed all the known diagnostic approaches and step by step exhaustively provide information about the main platelet function and structure analyses.

The article is written in a fluid way and it reads well. The paragraphs are well-structured and tables and figures are clear. The authors made a good work of bibliography and a good work of writing. I think that this article may be useful to the reader. I have only an observation regarding Figure 4: the below Legend of Figure 4 (double? Remove the upper one) needs to be revised.

Author Response

REPLY TO REVIEWER 2

Comments and Suggestions for Authors

In this Review, the Authors described the diagnostic algorithm to conduce to diagnosis of inherited platelet disorders.

My opinion regarding this article is positive. The authors comprehensively reviewed all the known diagnostic approaches and step by step exhaustively provide information about the main platelet function and structure analyses.

The article is written in a fluid way and it reads well. The paragraphs are well-structured and tables and figures are clear. The authors made a good work of bibliography and a good work of writing. I think that this article may be useful to the reader.

We would like to thank the reviewer for their very positive feedback on our manuscript.

 I have only an observation regarding Figure 4: the below Legend of Figure 4 (double? Remove the upper one) needs to be revised.

Thank you very much for the annotation. The text above the figure is not a caption. We have added 'The' to make this clear. We have also corrected the numbering of the letters in the figure legend, which was incorrect.

Reviewer 3 Report

Comments and Suggestions for Authors

In the review article entitled: „Diagnosis of Inherited Platelet Disorders: Clinical Evaluation, Functional and Molecular Assays” by Dr Sanchez-Fuentes et al., the Authors provided a very detailed and comprehensive review of scientific literature on inherited disorders of blood platelets. The review is well written, contains useful graphics (4 figures, 7 tables) however there are quite a few minor issues that should be corrected before the publication. The list of my minor comments is enclosed below.

  1. Table 2. Please give an explanation of all the crucial abbreviations presented in the table, such as IT, IPFD.
  2. Figure 3. PFA-100 was developed over 20 years ago. Now, there is also PFA-200 available. Besides, there are other analysers such as TTAS. The Figure 3, should be updated accordingly.
  3. Figure 3. There are many typo errors in this figure, e.g. secreatoin-> secretion, and/o -> and/or. I suggest to proofread a whole figure thoroughly.
  4. Figure 4. consists of panels A)-G). The explanation is given only for panels A)-F). It should be corrected.
  5. Table 6. First column. The concentrations of agonists missed proper units. It should be micromolar not molar concentrations for ADP, TRAP. There are also errors for other agonists.

Author Response

REPLY TO REVIEWER 3

Comments and Suggestions for Authors

In the review article entitled: „Diagnosis of Inherited Platelet Disorders: Clinical Evaluation, Functional and Molecular Assays” by Dr Sanchez-Fuentes et al., the Authors provided a very detailed and comprehensive review of scientific literature on inherited disorders of blood platelets. The review is well written, contains useful graphics (4 figures, 7 tables) however there are quite a few minor issues that should be corrected before the publication. The list of my minor comments is enclosed below.

First, we would like to thank the reviewer for the very positive feedback on our manuscript.

1. Table 2. Please give an explanation of all the crucial abbreviations presented in the table, such as IT, IPFD.

I am grateful for the suggestion. The most pertinent abbreviations have been explained

2. Figure 3. PFA-100 was developed over 20 years ago. Now, there is also PFA-200 available. Besides, there are other analysers such as TTAS. The Figure 3, should be updated accordingly.

Thank also for this comment. Indeed the PFA-200 is available and many lab are using it. Also the T-TAS systeme is being introduced in many laboratories, including our lab. We have updated the figure 3 as recommended

3. Figure 3. There are many typo errors in this figure, e.g. secreatoin-> secretion, and/o -> and/or. I suggest to proofread a whole figure thoroughly.

We are sorry for these mistakes, which have now been corrected.

4. Figure 4. consists of panels A)-G). The explanation is given only for panels A)-F). It should be corrected.

We have corrected the numbering of the letters in the figure legend, which was incorrect.

5. Table 6. First column. The concentrations of agonists missed proper units. It should be micromolar not molar concentrations for ADP, TRAP. There are also errors for other agonists.

We have updated the table and it looks good now.